# Cumulative Fluid Balance during Extracorporeal Membrane Oxygenation and Mortality in Patients with Acute Respiratory Distress Syndrome

**DOI:** 10.3390/membranes11080567

**Published:** 2021-07-28

**Authors:** Li-Chung Chiu, Li-Pang Chuang, Shih-Wei Lin, Yu-Ching Chiou, Hsin-Hsien Li, Yung-Chang Chen, Yu-Jr Lin, Chee-Jen Chang, Feng-Chun Tsai, Ko-Wei Chang, Han-Chung Hu, Chung-Chi Huang, Shaw-Woei Leu, Kuo-Chin Kao

**Affiliations:** 1Department of Thoracic Medicine, Chang Gung Memorial Hospital, Chang Gung University College of Medicine, Taoyuan 33305, Taiwan; pomd54@cgmh.org.tw (L.-C.C.); r5243@adm.cgmh.org.tw (L.-P.C.); ec108146@adm.cgmh.org.tw (S.-W.L.); b9302072@cgmh.org.tw (K.-W.C.); h3226@cgmh.org.tw (H.-C.H.); cch4848@cgmh.org.tw (C.-C.H.); kck0502@cgmh.org.tw (K.-C.K.); 2Graduate Institute of Clinical Medical Sciences, College of Medicine, Chang Gung University, Taoyuan 33302, Taiwan; 3Department of Thoracic Medicine, New Taipei Municipal TuCheng Hospital and Chang Gung University, Taoyuan 33302, Taiwan; 4School of Medicine, Chang Gung University College of Medicine, Taoyuan 33302, Taiwan; sayatsukimi@gmail.com; 5Department of Respiratory Therapy, Chang Gung University College of Medicine, Taoyuan 33302, Taiwan; hsinhsien@mail.cgu.edu.tw; 6Institute of Emergency and Critical Care Medicine, School of Medicine, National Yang Ming Chiao Tung University, Taipei 11221, Taiwan; 7Kidney Research Center, Department of Nephrology, Chang Gung Memorial Hospital, Chang Gung University College of Medicine, Taoyuan 33305, Taiwan; cyc2356@gmail.com; 8Research Services Center for Health Information, Chang Gung University, Taoyuan 33302, Taiwan; doublelin15@gmail.com (Y.-J.L.); cjchang@mail.cgu.edu.tw (C.-J.C.); 9Clinical Informatics and Medical Statistics Research Center, Chang Gung University, Taoyuan 33302, Taiwan; 10Division of Cardiovascular Surgery, Chang Gung Memorial Hospital, Taoyuan 33305, Taiwan; lutony@cgmh.org.tw; 11Department of Respiratory Therapy, Chang Gung Memorial Hospital, Chang Gung University College of Medicine, Taoyuan 33305, Taiwan

**Keywords:** acute respiratory distress syndrome, extracorporeal membrane oxygenation, cumulative fluid balance, mortality

## Abstract

Extracorporeal membrane oxygenation (ECMO) is considered a salvage therapy in cases of severe acute respiratory distress syndrome (ARDS) with profound hypoxemia. However, the need for high-volume fluid resuscitation and blood transfusions after ECMO initiation introduces a risk of fluid overload. Positive fluid balance is associated with mortality in critically ill patients, and conservative fluid management for ARDS patients has been shown to shorten both the duration of mechanical ventilation and time spent in intensive care, albeit without a significant effect on survival. Nonetheless, few studies have addressed the influence of fluid balance on clinical outcomes in severe ARDS patients undergoing ECMO. In the current retrospective study, we examined the impact of cumulative fluid balance (CFB) on hospital mortality in 152 cases of severe ARDS treated using ECMO. Overall hospital mortality was 53.3%, and we observed a stepwise positive correlation between CFB and the risk of death. Cox regression models revealed that CFB during the first 3 days of ECMO was independently associated with higher hospital mortality (adjusted hazard ratio 1.110 [95% CI 1.027–1.201]; *p* = 0.009). Our findings indicate the benefits of a conservative treatment approach to avoid fluid overload during the early phase of ECMO when dealing with severe ARDS patients.

## 1. Introduction

Extracorporeal membrane oxygenation (ECMO) is considered a rescue therapy for life-threatening hypoxemia in patients with severe acute respiratory distress syndrome (ARDS) [1,2,3,4]. Oxygen delivery during ECMO depends on the flow of blood through the circuit, which is limited primarily by the size of the drainage cannula, the diffusion properties of the membrane oxygenator and the concentration of hemoglobin. Increasing hemoglobin concentrations via red blood cell transfusion is commonly used to maintain adequate oxygen delivery during ECMO [1,2,3,4,5,6]. During the initial phase of ECMO, high-volume fluid resuscitation and blood transfusions are often required to ensure sufficient blood flow and to correct anemia or coagulopathy [1,6,7]. However, difficulties in maintaining normal extracellular fluid volumes can lead to fluid overload with corresponding effects on clinical outcomes [7].

Positive fluid balance is associated with a higher risk of death in cases of sepsis and critically ill patients [8,9,10,11,12]. In ARDS patients, optimal fluid management should provide adequate oxygen delivery to the body and avoid increases in lung edema, which could further impair gas exchange [13]. One randomized study reported that the use of a conservative fluid strategy in patients with ARDS could improve lung function and shorten the duration of mechanical ventilation and time spent in intensive care, albeit without a significant influence on 60-day mortality [14]. One systemic review and meta-analysis revealed that a conservative fluid strategy in cases of ARDS increased the number of ventilator-free days and decreased the length of stays in the intensive care unit (ICU) but without significant benefits pertaining to mortality [15].

ARDS patients who are not in shock should be treated using a conservative fluid strategy; however, fluid management in severe ARDS patients undergoing ECMO remains controversial with few published reports. Previous studies have reported that early positive fluid balance was independently associated with mortality in patients undergoing ECMO support [16,17]; however, none of those studies focused on the subgroup of patients with severe ARDS receiving ECMO. Our objective in the current study was to investigate the effect of cumulative fluid balance (CFB) during the early phase of ECMO on clinical outcomes and hospital mortality in patients with severe ARDS.

## 2. Materials and Methods

### 2.1. Study Design and Patients

This study focused on patients with severe ARDS who underwent ECMO between May 2006 and October 2015 in medical and surgical ICUs at a tertiary-care referral facility in Taiwan (Chang Gung Memorial Hospital (CGMH); 3700-bed general ward; 278-bed adult ICU). Exclusion criteria were as follows: (1) age < 20 years, (2) malignancies with poor prognosis within 5 years, (3) significant underlying comorbidities or severe multiple organ failure refractory to treatment, (4) end-stage renal disease, and (5) mortality within 3 days after ECMO initiation. At our facility, the decision to initiate ECMO cannulation is made by the treating intensivist in consultation with a cardiac surgeon, whereas the decision to initiate renal replacement therapy (RRT) is made by the treating intensivist in consultation with a nephrologist. During this study period, the criterion for ECMO initiation in severe ARDS patients was persistent hypoxemia (PaO_2_/FiO_2_ ratio <80 mmHg) for at least 6 h, despite aggressive mechanical ventilation support such as positive end-expiratory pressure (PEEP) >10 cm H_2_O or peak inspiratory pressure >35 cm H_2_O. All severe ARDS patients with ECMO support were deeply sedated and paralyzed with continuous neuromuscular blockade during the initial phase of ECMO support, and respiratory mechanics data were collected during neuromuscular blockade. Based on experience, the criteria for weaning from ECMO were resolving lung infiltration, lung compliance > 20 mL/cm H_2_O, PaO_2_ > 60 mmHg as well as PaCO_2_ < 45 mmHg under FiO_2_ ≦ 0.4, PEEP ≦ 6–8 cm H_2_O, and peak inspiratory pressure ≦ 30 cm H_2_O. The local Institutional Review Board for Human Research approved this study (CGMH IRB No. 201600632B0), and the need for informed consent was waived due to the retrospective nature of our analysis.

### 2.2. Definitions

ARDS was defined in accordance with the Berlin criteria [18]. Mechanical power was calculated using the following equation [19]:Mechanical power (Joules/minutes) (J/min) = 0.098 × tidal volume × respiratory rate × (peak inspiratory pressure − 1/2 × driving pressure).

Acute kidney injury was defined according to the KDIGO (Kidney Disease: Improving Global Outcomes) classification system and staged for severity based on serum creatinine or urine output [20].

CFB (mL) was defined as cumulative total fluid input minus cumulative total fluid output.

### 2.3. Data Collection

Demographic data, etiologies of ARDS, underlying comorbidities, Sequential Organ Failure Assessment (SOFA) score, and lung injury score were collected prior to ECMO initiation. The dates of hospital and ICU admission, mechanical ventilator initiation and liberation, ECMO cannulation and decannulation, RRT, ICU and hospital discharge, and time of death were recorded.

Fluid balance was calculated from total fluid intake (enteral fluids, intravenous fluids, blood products, and RRT replacement fluids) and output (urine output, enteral loss, drainage fluids, effluent dialysate from RRT, and insensible fluid loss), both of which were recorded daily throughout the ICU stay. Patients were stratified according to the quartiles of CFB during the first 3 days after ECMO support with the aim of elucidating the influence of early fluid balance during ECMO on clinical outcomes. Arterial blood gas parameters and mechanical ventilator settings were recorded at the time of ECMO initiation and at approximately 10 a.m. on days 1, 2, and 3 after ECMO initiation.

### 2.4. ECMO Systems

The ECMO circuit consisted of a centrifugal pump and hollow-fiber microporous membrane oxygenator with heparin-bound Carmeda BioActive Surface (Carmeda, a subsidiary of WL Gore and Associates, Flagstaff, AZ, USA) using Capiox emergent bypass system (Terumo, Tokyo, Japan). We used two wire-wound polyurethane vascular cannulas (DLP Medtronic, Minneapolis, MN, USA (inflow, 19F to 23F, and outflow, 17F to 21F)), and the femoral–jugular venovenous (VV) ECMO was established through percutaneous cannulation. Patients would be shifted to venoarterial (VA) mode ECMO if encountered with cardiopulmonary cerebral resuscitation, cardiac arrest, or pulmonary hypertension. ECMO mode was soon changed to VV ECMO once the hemodynamics of ARDS patients stabilized.

The ECMO gas flow rate was set high initially (10 L/min, pure oxygen), and the blood pump speed was gradually increased to achieve optimal oxygen saturation (90% or more). Modest volume replacement was necessary initially to improve unsteady ECMO blood flow and oxygenation. Red blood cell transfusion was performed to maintain hemoglobin at least above 10 g/dL. After optimizing the ECMO, it was crucial to remove the excessive extravascular lung water (i.e., no systemic edema, within 5% of dry weight) with diuretics or continuous renal replacement therapy to improve lung function and compliance. The hourly fluid balance goal was set at approximately −100 mL/h and modulated according to dry weight to achieve negative fluid balance, and extrapulmonary organ function was closely monitored.

### 2.5. Statistical Analysis

Continuous variables are presented as mean ± standard deviation or median (interquartile range), and categorical variables are reported as numbers (percentages). Analysis of variance, the Kruskal–Wallis test, a Student’s *t*-test, or the Mann–Whitney *U* test was used to compare continuous variables among groups. Categorical variables were tested using the chi-square test for equal proportions or Fisher’s exact test. Receiver operating characteristic curves and the Youden index were used to determine the cutoff to dichotomize continuous variables. Risk factors associated with hospital mortality were analyzed using univariate analysis, followed by a Cox proportional hazard regression model with stepwise selection. The results are presented in terms of hazard ratio (HR) and 95% confidence interval (CI). CFB was modeled as a continuous variable (per 1-L increase), and the quartiles of CFB were modeled as categorical variables with the first quartile as the reference category. Cumulative mortality curves were generated as a function of time using the Kaplan–Meier approach and compared using the log-rank test. All statistical analyses were performed using SPSS 22.0 statistical software, and a two-sided *p* value < 0.05 was considered statistically significant.

## 3. Results

### 3.1. Patients

During the study period, a total of 192 patients with severe respiratory failure receiving ECMO were included. After excluding 40 patients, 152 patients were in the final analysis, which focused on the impact of CFB during the first 3 days of ECMO on hospital mortality (Figure 1). Mortality rates were as follows: overall (53.3%), positive cumulative fluid balance (n = 98; 62.2%), and negative cumulative fluid balance (n = 54; 37%) (*p* < 0.001). The likelihood of hospital mortality was higher among patients in higher CFB quartiles. The ECMO techniques did not show significant difference, and a unified ECMO system was used during the study period. Patients in the later years (2012–2015) received significantly lower tidal volume, higher PEEP, lower peak inspiratory pressure, and lower mechanical power during the first 3 days of ECMO than did patients in the earlier years (2006–2011) (Appendix A). However, hospital mortality was not significantly different between patients in the earlier years and later years of the study period (2006–2011: 77 patients, mortality rate 54.5%; 2012–2015: 75 patients, mortality rate 52%, *p* = 0.753).

### 3.2. Comparison of Survivors and Nonsurvivors

As shown in Table 1, nonsurvivors tended to be older than survivors. We did not observe a significant difference between survivors and nonsurvivors in terms of fluid balance prior to ECMO; however, nonsurvivors presented a significantly higher CFB balance at 24 h and at 3 days after ECMO. Fluid balance was more positive after ECMO support from day 1 to day 7 in nonsurvivors than in survivors. Mean fluid balance in survivors changed to negative at day 3 after ECMO initiation, whereas mean fluid balance in nonsurvivors changed to negative at day 5 after ECMO initiation (Figure 2). Nonsurvivors received VA ECMO implantation more often than did survivors. In terms of ventilator settings during ECMO, nonsurvivors received significantly higher mechanical power, higher peak inspiratory pressure, and lower dynamic compliance than did survivors (all *p* < 0.05). During ECMO, nonsurvivors required more inotropic support and more frequent RRT with higher SOFA scores (all *p* < 0.05). The most common indications for RRT were fluid overload (n = 70, 80%), followed by refractory hyperkalemia (n = 8, 9%), uremia (n = 7, 8%), and severe metabolic acidosis (n = 3, 3%).

### 3.3. Comparisons of Cumulative Fluid Balance at 3 Days after ECMO

The value of CFB for each quartile was as follows: quartile 1 (<−873 mL), quartile 2 (−873 to 1190 mL), quartile 3 (1190 to 3935 mL), and quartile 4 (> 3935 mL). As shown in Table 2, there were no significant differences among the CFB quartiles in terms of age or body mass index. We observed no significant differences among the groups in terms of SOFA scores prior to ECMO initiation; however, SOFA scores during ECMO support were significantly higher in higher CFB quartiles (*p* = 0.041). We observed no significant differences among the groups in terms of oxygenation status (i.e., PaO_2_/FiO_2_) or ventilator settings during ECMO, except for lower dynamic compliance in CFB higher quartiles (*p* = 0.005). Urine outputs were lowest and the need for RRT were highest in the CFB quartile 4. The likelihood of hospital mortality was also higher among patients in higher CFB quartiles (*p* = 0.009).

### 3.4. Outcomes

As shown in Table 3, there was a stepwise increase in hospital mortality corresponding to an increase in CFB quartile, with significant between-group differences in terms of 28-, 60-, and 90-day hospital mortality (all *p* < 0.05). We observed no significant differences among the quartiles in terms of ECMO duration, mechanical ventilator duration, length of ICU stay, or length of hospital stay. Patients in lower CFB quartiles presented more ECMO-free days by day 28; however, the effect was not significant. We observed significantly higher 28- and 60-day ventilator-free days in lower CFB quartiles (*p* = 0.002 and 0.001, respectively). We also observed significantly lower overall 90-day survival rates in quartile 4 (overall comparison, *p* = 0.001, log-rank test), as follows: quartile 1 (63.2%), quartile 2 (55.3%), quartile 3 (50%), and quartile 4 (31.6%). We observed no significant differences in overall 90-day survival rates among quartiles 1, 2, or 3 (Figure 3).

### 3.5. Factors Associated with Hospital Mortality

After adjusting for significant confounding variables, Cox proportional hazard regression models revealed that CFB during the first 3 days of ECMO was independently associated with an increased risk of death (adjusted HR 1.110 [95% CI 1.027–1.201]; *p* = 0.009) when CFB was considered as a continuous variable. When the quartiles of CFB were considered as categorical variables with the first quartile as the reference category, the risk of death revealed a stepwise increasing trend with an increase in CFB quartile. The risk of death in quartile 4 was significantly higher than in quartile 1 (adjusted HR 2.710 [95% CI 1.379–5.325]; *p* = 0.004); however, differences among quartiles 1, 2, and 3 did not reach the level of significance (Table 4).

### 3.6. Comparisons of VV-ECMO- and VA-ECMO-Supported ARDS Patients

As shown in Table 5, VA ECMO patients had a lower percentage of diabetes mellitus. The SOFA score was significantly higher in VA ECMO patients before ECMO. In terms of ventilator settings during ECMO, VA ECMO cases received significantly higher mechanical power, higher tidal volume, lower PEEP, and higher peak inspiratory pressure than did VV ECMO cases (all *p* < 0.05). We did not observe a significant difference between VV ECMO and VA ECMO cases in terms of fluid balance before and during ECMO. No significant differences were observed between the two groups in terms of SOFA score, acute kidney injury development, diuretic use, inotropic support, and RRT use during ECMO. VA ECMO patients were associated with significantly higher hospital mortality than VV ECMO patients (*p* = 0.020).

## 4. Discussion

The primary insight gained in this research was the fact that excessive CFB during the first 3 days of ECMO was independently associated with an increased risk of death in severe ARDS patients. We also observed higher mortality rates in higher CFB quartiles.

Most severe ARDS patients receive VV ECMO for respiratory support, and there is no direct hemodynamic support provided by the extracorporeal circuit. However, unstable hemodynamics is frequently encountered during the early phase of ECMO, and it is necessary to add volume for ECMO enforcement. Hemodynamics are managed with fluid or blood volume replacement and inotropes in the same fashion as in patients without VV ECMO support [1]. Bleeding remains a clinically significant ECMO complication by anticoagulation, and blood transfusions are needed to correct anemia or coagulopathy.

The blood volume during ECMO should be maintained at a level high enough to keep right atrial pressure in the range of 5–10 mmHg. This will assure adequate volume for venous drainage, as long as the resistance of the drainage cannula is appropriate. Echocardiography is an excellent tool to assess hemodynamic function and help guide management during VV ECMO. If cardiac failure occurs during VV ECMO support, the patients could be converted to VA ECMO. If the systemic perfusion pressure is inadequate (i.e., low urine output or poor perfusion), pressure can be increased by adding blood volume through transfusion or by using low doses of vasopressors. If systemic oxygen delivery is not adequate (i.e., venous saturation less than 70%), the pump flow should be increased until perfusion is adequate. If blood volume is required to gain adequate flow, the relative advantages of using blood products and crystalloid solutions should be considered [7].

Researchers have yet to establish optimal fluid balance targets beyond the initial resuscitation period for patients treated with ECMO (including severe ARDS patients). In guidelines published by the Extracorporeal Life Support Organization, the goal of fluid management in patients with ECMO should be to maintain normal extracellular fluid volume (i.e., dry weight), adequate hematocrit to optimize oxygen delivery, normal body weight without fluid overload, and normal blood volume [7]. However, acute inflammatory reactions during the early phase of ECMO support can increase capillary leakage with increased vascular permeability and tissue edema and may lead to eventual organ injury [21]. Furthermore, this situation can be exacerbated by excessive crystalloid infusion, which increases the likelihood of fluid overload. This means that practitioners should implement a conservative fluid management strategy aimed at achieving negative fluid balance via aggressive diuresis and continuous hemofiltration when necessary [1,7].

During the early phase of ECMO, fluid overload can profoundly affect outcomes. One previous report concluded that early positive fluid balance (particularly at day 3 of VV or VA ECMO) is an independent predictor of 90-day mortality among adult patients [16]. Another retrospective study found that excessive CFB during the first 3 days of ECMO increased the risk of mortality, regardless of whether the patients were suffering from cardiovascular or pulmonary diseases and whether VV or VA ECMO was use [17]. Note, however, that those studies included all patients requiring ECMO for circulatory or pulmonary failure, including patients with severe ARDS (i.e., n = 53 and 75, respectively). In other words, those studies did not evaluate the effect of early fluid status during ECMO on clinical outcomes exclusively among severe ARDS patients. Furthermore, neither study investigated the correlation between early fluid status during ECMO and clinical or ventilatory variables in severe ARDS patients. In the current study, focusing exclusively on severe ARDS patients receiving ECMO (n = 152), fluid status was not significantly different between VV and VA ECMO patients before and during ECMO. However, CFB during the first 3 days of ECMO was independently associated with hospital mortality. We also observed a stepwise increase in hospital mortality corresponding to higher CFB quartiles.

Fluid overload brings with it an elevated risk of pulmonary edema, which can have profound effects on oxygenation and lung compliance with a corresponding increase in respiratory work and difficulties pertaining to ventilator weaning [13,14,22,23]. We did not observe a significant difference between CFB quartiles in terms of oxygenation status (i.e., PaO_2_/FiO_2_) during ECMO. Lung compliance is a prerequisite for weaning from ECMO [1,3,4]; however, we observed significantly lower dynamic compliance in higher CFB quartiles (i.e., lowest in CFB quartile 4). We also observed that patients in higher CFB quartiles had fewer ECMO-free days at day 28, though the effect was not significant. Taken together, these results indicate that fluid overload could influence lung compliance, delay weaning, and increase ECMO duration [7,24]. The number of ventilator-free days at day 28 and day 60 was significantly lower in higher CFB quartiles, indicating that early fluid status during ECMO may also affect the weaning of severe ARDS patients from mechanical ventilators. Finally, mechanical ventilator settings were associated with mortality in ARDS patients undergoing ECMO [2,3,4], and VA ECMO patients received higher airway pressures and higher tidal volume during ECMO than VV ECMO patients, which may be associated with significantly higher hospital mortality in VA ECMO patients.

Critically ill patients undergoing ECMO face a higher risk of developing acute kidney injury, which may predispose them to fluid overload and impaired oxygen transport, thereby necessitating RRT and increasing the risk of death [2,16,24,25,26]. The factors contributing to acute kidney injury during ECMO included premorbid comorbidities and acute inflammatory and immune-mediated processes, related to the exposure of blood to the extracorporeal circulation, hemodynamic instability, hypercoagulable state, hemolysis, and exposure to nephrotoxic substances [24,25]. One recent international multicenter cohort reported that the need for RRT in severe ARDS patients receiving ECMO was independently associated with 6-month mortality [26]. In our cohort, the strongest indication for RRT during ECMO was fluid overload (70 patients; 80%). Furthermore, RRT itself was a strong indicator of mortality, as indicated by the fact that the incidence of fluid accumulation requiring RRT was higher among nonsurvivors than among survivors. Note also that the highest likelihood of requiring RRT and the highest mortality were both observed in CFB quartile 4. In terms of renal function (i.e., serum creatinine), we did not observe significant differences between survivors and nonsurvivors before and during ECMO or among patients in different CFB quartiles during ECMO. Nonetheless, the initiation of RRT may be an indicator of multiple organ failure, which is no doubt associated with poor outcomes [8,24].

The most common cause of death among ARDS patients is multiorgan failure [22]. One international multicenter study reported that for severe ARDS patients, extrapulmonary organ failure during ECMO had a significantly negative impact on 6-month mortality [26]. Our findings revealed no significant differences between survivors and nonsurvivors in terms of fluid balance or SOFA score prior to ECMO initiation; however, fluid balance and SOFA scores during the first 3 days of ECMO were significantly higher among nonsurvivors than among survivors. We also observed higher SOFA scores during ECMO in patients with higher CFB quartiles, which may reflect higher severity in illness and organ dysfunction. Excess fluid accumulation has been shown to exacerbate tissue edema, stretch vascular walls, worsen vascular permeability, and eventually promote organ dysfunction with corresponding effects on clinical outcomes [8,11,12,23,27]. Nonetheless, determining the causal relationship between fluid overload and organ dysfunction was difficult due to the retrospective nature of our study.

This study was hindered by a number of limitations. First, our focus on patients receiving ECMO for severe ARDS limited the number of patients in the CFB quartiles. Furthermore, this was a retrospective study conducted in a single tertiary-care referral facility. These facts no doubt limit the generalizability of our findings. Second, we made no attempt to assess the type or amount of fluid administration (e.g., colloids, crystalloids, or blood products), inotrope dosage, diuretic dosage, and RRT modalities’ dose and frequency. Furthermore, we did not analyze hemodynamic parameters as an indicator of fluid status. Third, we did not analyze fluid output other than urine. Thus, we were unable to determine the effects of limiting excess fluid intake or promoting fluid output within the context of clinical outcomes. Note, however, that the definition of fluid overload has yet to be clearly established, and researchers have yet to determine the optimal time point at which CFB should be assessed to predict outcomes. Finally, our objective in this observational study was to evaluate the correlation between CFB and mortality during the early phase of ECMO, without considering issues pertaining to causality.

## 5. Conclusions

Our findings revealed that CFB during the first 3 days of ECMO was independently associated with 90-day hospital mortality, as indicated by higher mortality rates corresponding to higher CFB quartiles. These results indicate the need for a conservative fluid strategy to prevent fluid overload during the early phase of ECMO when dealing with severe ARDS patients.

## Figures and Tables

**Figure 1 membranes-11-00567-f001:**
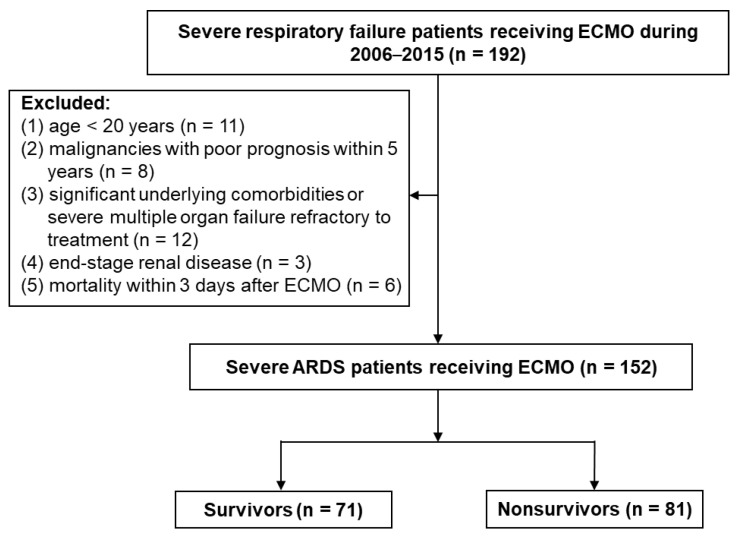
Flowchart of enrolling patients with severe ARDS with ECMO support. ARDS, acute respiratory distress syndrome; ECMO, extracorporeal membrane oxygenation.

**Figure 2 membranes-11-00567-f002:**
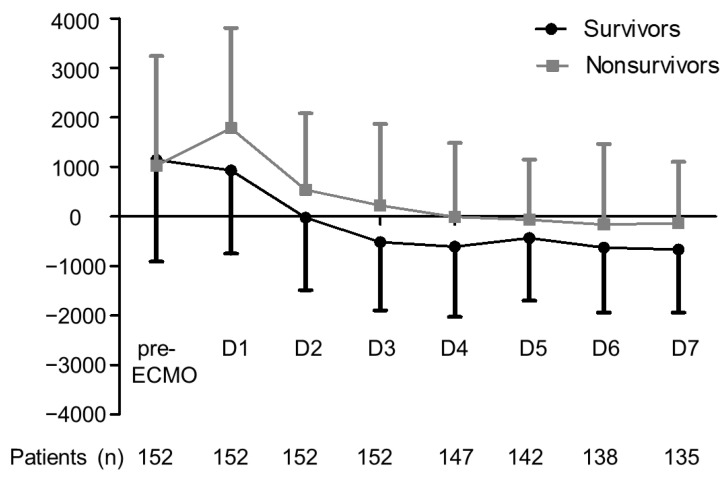
Mean fluid balance in survivors and nonsurvivors of severe ARDS during the first week of ECMO support. Error bars represent the mean ± standard deviation. Dark line denotes survivors, and gray line denotes nonsurvivors. * *p* < 0.05 comparing survivors and nonsurvivors. ARDS, acute respiratory distress syndrome; ECMO, extracorporeal membrane oxygenation.

**Figure 3 membranes-11-00567-f003:**
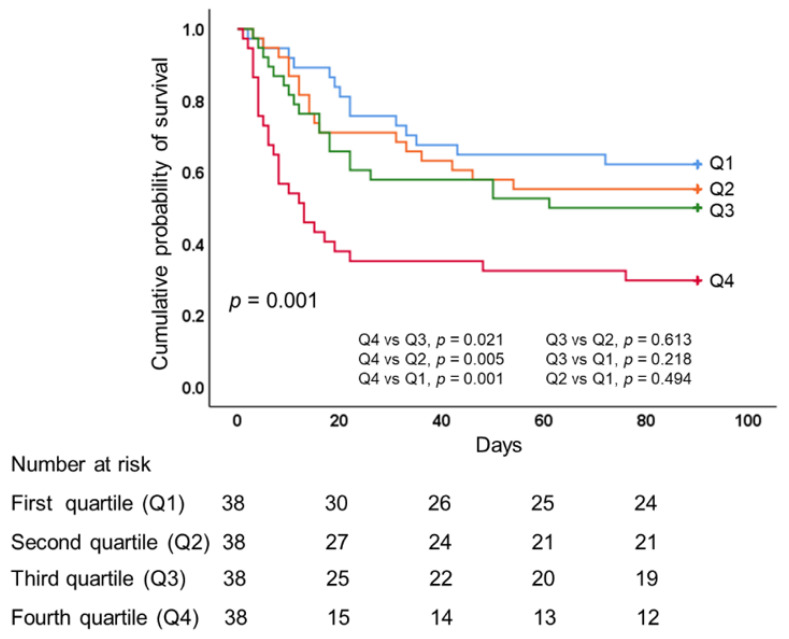
Kaplan–Meier 90-day survival curves of patients undergoing ECMO for severe acute respiratory distress syndrome, as stratified by quartiles of cumulative fluid balance during the first 3 days of ECMO. ECMO, extracorporeal membrane oxygenation.

**Table 1 membranes-11-00567-t001:** Background characteristics and clinical variables: survivors and nonsurvivors.

Variables		All	Survivors	Nonsurvivors	*p*
		(n = 152)	(n = 71)	(n = 81)	
	Age (years)	50.3 ± 16.4	46.0 ± 16.5	54.1 ± 15.4	0.002
	Male (gender)	103 (67.8%)	48 (67.6%)	55 (67.9%)	0.969
	Body mass index (kg/m^2^)	25.8 ± 5.3	26.0 ± 5.8	25.6 ± 4.7	0.631
ARDS etiologies					
	Pulmonary cause	118 (78%)	59 (83%)	59 (73%)	0.130
	Extrapulmonary cause	34 (22%)	12 (17%)	22 (27%)	0.130
	Diabetes mellitus	40 (26%)	23 (32%)	17 (21%)	0.111
	Chronic liver disease	21 (14%)	6 (9%)	15 (19%)	0.073
	Immunocompromised status	40 (26%)	11 (16%)	29 (36%)	0.005
	Chronic kidney disease	17 (11.2%)	8 (11.2%)	9 (11.1%)	0.976
	SOFA score before ECMO	10.8 ± 3.2	10.3 ± 3.1	11.3 ± 3.2	0.067
	Lung injury score before ECMO	3.4 ± 0.4	3.4 ± 0.4	3.3 ± 0.4	0.106
	PaO_2_/FiO_2_ (mmHg) before ECMO	63 (52–88)	64 (53–80)	63 (52–107)	0.168
	ECMO VA mode	24 (15.8%)	6 (8.5%)	18 (22.2%)	0.020
	ECMO blood flow rate (L/min) at day 1	3.4 ± 0.7	3.3 ± 0.8	3.5 ± 0.7	0.079
	ECMO blood flow rate (L/min) at day 3	3.8 ± 0.8	3.7 ± 1.0	3.9 ± 0.7	0.269
	SOFA score from day 1 to day 3 on ECMO	9.6 ± 2.3	8.8 ± 1.9	10.4 ± 2.4	<0.001
	PaO_2_/FiO_2_ (mmHg) from day 1 to day 3 on ECMO	178 (131–240)	200 (146–247)	165 (124–211)	0.588
Ventilator settings from day 1 to day 3 on ECMO					
	Mechanical power (J/min)	12.1 ± 6.2	10.9 ± 4.3	13.1 ± 7.4	0.022
	Tidal volume (mL/kg PBW)	6.0 ± 2.2	6.1 ± 2.0	6.0 ± 2.4	0.914
	PEEP (cm H_2_O)	12.0 ± 3.3	12.3 ± 3.2	11.7 ± 3.3	0.202
	Peak inspiratory pressure (cm H_2_O)	31.7 ± 5.6	30.6 ± 5.1	32.8 ± 5.9	0.018
	Mean airway pressure (cm H_2_O)	17.7 ± 4.0	17.4± 3.6	17.9 ± 4.3	0.406
	Dynamic compliance (mL/cm H_2_O)	19.2 ± 8.1	21.1 ± 7.7	17.4 ± 8.1	0.006
Fluid balance, mL					
	Before ECMO 24 h	923 (−258 to 2125)	1027 (−287 to 2341)	795 (−193 to 1781)	0.751
	First 24 h	1327 (57 to 2800)	846 (−160 to 2095)	1688 (219 to 3668)	0.006
	Cumulative 3 days	1190 (−873 to 3935)	277 (−1798 to 2384)	1927 (−100 to 5266)	<0.001
Total fluid input, mL					
	Before ECMO 24 h	3007 (2255–4117)	2772 (2281–3924)	3437 (2255–4304)	0.597
	First 24 h	4774 (3545–5926)	4065 (3118–5441)	5213 (4010–6010)	0.083
	Cumulative 3 days	13,013 (9996–17,095)	12,675 (10,039–15,056)	13,374 (9871–17,626)	0.841
Urine output, mL					
	Before ECMO 24 h	1500 (550–2380)	1585 (700–2750)	1350 (475–2093)	0.054
	First 24 h	1318 (396–2536)	1642 (750–2705)	950 (183–2438)	0.055
	Cumulative 3 days	4890 (1375–8758)	6853 (1731–9779)	3760 (681–7530)	0.042
Blood product transfusions, mL					
	Red blood cell	505 (390–1120)	520 (395–1075)	430 (385–1180)	0.803
	Fresh frozen plasma	560 (280–1120)	490 (280–1100)	590 (280–1120)	0.800
	Platelet concentrate	250 (130–560)	230 (125–550)	325 (198–573)	0.721
Hemoglobin (g/dL)					
	Before ECMO	10.7 ± 2.4	11.5 ± 2.4	10.0 ± 2.3	0.008
	Day 1	10.7 ± 2.9	11.4 ± 3.4	9.9 ± 2.1	0.004
	Day 3	10.2 ± 1.6	10.2 ± 1.8	10.1 ± 1.3	0.881
Serum creatinine (mg/dL)					
	Before ECMO	2.0 ± 2.1	2.0 ± 2.4	2.0 ± 1.9	0.946
	Day 1	2.1 ± 1.8	2.1 ± 1.9	2.1 ± 1.8	0.782
	Day 3	2.0 ± 1.5	1.8 ± 1.7	2.1 ± 1.3	0.343
AKI during ECMO (n)					
	Stage 1	62 (40.8%)	29 (40.8%)	33 (40.7%)	0.990
	Stage 2	31 (20.4%)	10 (14.1%)	21 (25.9%)	0.071
	Stage 3	20 (13.2%)	7 (9.9%)	13 (16%)	0.260
	Diuretics (n)	106 (69.7%)	52 (73.2%)	54 (66.7%)	0.379
	Inotropes (n)	133 (87.5%)	53 (74.6%)	80 (98.8%)	<0.001
	RRT (n)	88 (58%)	30 (42%)	58 (72%)	<0.001
	RRT initiation prior to ECMO (n)	15 (9.9%)	4 (5.6%)	11 (13.6%)	0.112
	Time from RRT to ECMO (days)	5 (2–18)	2 (1–2)	11 (5–20)	0.002
	RRT initiation during ECMO (n)	73 (48%)	26 (36.7%)	47 (58%)	0.008
	Time from ECMO to RRT (days)	1 (0–3)	0 (0–1)	1 (0–5)	0.004
	Duration of ECMO (days)	9 (6–15)	9 (5–13)	11 (6–16)	0.089
	Duration of RRT (days)	13 (4–30)	16 (5–34)	11 (4–28)	0.163

Data are presented as mean ± standard deviation, count, or median (interquartile range). Abbreviations: AKI, acute kidney injury; ARDS, acute respiratory distress syndrome; ECMO, extracorporeal membrane oxygenation; FiO_2_, fraction of inspired oxygen; PaO_2_, partial pressure of oxygen in arterial blood; PBW, predicted body weight; PEEP, positive end-expiratory pressure; RRT, renal replacement therapy; SOFA, Sequential Organ Failure Assessment; VA, venoarterial.

**Table 2 membranes-11-00567-t002:** Background characteristics and clinical variables as a function of quartiles of cumulative fluid balance at 3 days after ECMO.

Variables		First Quartile	Second Quartile	Third Quartile	Fourth Quartile	*p*
		(n = 38)	(n = 38)	(n = 38)	(n = 38)	
	Age (years)	48.1 ± 16.9	51.0 ± 15.7	46.3 ± 16.0	55.3 ± 16.1	0.088
	Male (gender)	27 (71.1%)	28 (73.7%)	27 (71.1%)	21 (55.3%)	0.347
	Body mass index (kg/m^2^)	26.7 ± 6.6	26.4 ± 4.1	26.3 ± 5.3	23.8 ± 4.5	0.110
	SOFA score before ECMO	9.9 ± 3.3	11.6 ± 2.7	10.9 ± 3.2	10.8 ± 3.3	0.153
	PaO_2_/FiO_2_ (mmHg) before ECMO	58 (52–96)	68 (56–87)	57 (48–86)	64 (54–95)	0.998
	ECMO VA mode	6 (15.8%)	7 (18.4%)	5 (13.2%)	6 (15.8%)	0.941
	ECMO blood flow rate (L/min) at day 1	3.4 ± 0.8	3.6 ± 0.6	3.3 ± 0.6	3.4 ± 0.8	0.570
	ECMO blood flow rate (L/min) at day 3	3.9 ± 0.8	4.1 ± 0.4	3.8 ± 1.1	3.7 ± 0.8	0.776
	SOFA score from day 1 to day 3 on ECMO	9.0 ± 2.3	9.1 ± 2.0	9.6 ± 2.5	10.5 ± 2.2	0.041
	PaO_2_/FiO_2_ (mmHg) from day 1 to day 3 on ECMO	167 (127–275)	160 (127–241)	192 (154–231)	169 (130–222)	0.696
Ventilator settings from day 1 to day 3 on ECMO						
	Mechanical power (J/min)	12.3 ± 6.0	13.0 ± 6.2	11 ± 4.7	12.1 ± 7.9	0.589
	Tidal volume (mL/kg PBW)	6.7 ± 2.6	6.0 ± 2.1	5.9 ± 1.7	5.5 ± 2.3	0.135
	PEEP (cm H_2_O)	11.9 ± 3.2	11.7 ± 3.5	12.4 ± 3.3	12 ± 3.1	0.839
	Peak inspiratory pressure (cm H_2_O)	30.5 ± 5.0	31.8 ± 5.7	31.8 ± 4.8	33.6 ± 6.2	0.101
	Mean airway pressure (cm H_2_O)	16.7 ± 3.8	17.5 ± 4.5	17.5 ± 3.4	19.0 ± 4.1	0.088
	Dynamic compliance (mL/cm H_2_O)	22.1 ± 7.8	19.6 ± 8.2	19.3 ± 7.5	15.6 ± 7.1	0.005
Fluid balance, mL						
	Before ECMO 24 h	494 (−440 to 1537)	363 (−324 to 1484)	1102 (221 to 2244)	1242 (−28 to 3118)	0.024
	First 24 h	−376 (−1327 to 267)	645 (57 to 2035)	1539 (595 to 2427)	3646 (2407 to 4379)	<0.001
	Cumulative 3 days	−2158 (−3844 to −1644)	223 (−486 to 713)	2179 (1797to 2914)	5995 (4556 to 8128)	<0.001
Total fluid input, mL						
	Before ECMO 24 h	2760 (1856–3970)	3700 (2745–4227)	3025 (2230–4330)	2980 (1870–4836)	0.405
	First 24 h	4065 (3158–5213)	4047 (3350–5673)	4696 (3545–5499)	5795 (4800–7351)	0.095
	Cumulative 3 days	12,023 (9405–15,900)	12,899 (9598–15,057)	12,774 (10,088–16,847)	14,654 (11,365–18,983)	0.717
Urine output, mL						
	Before ECMO 24 h	2300 (1008–3565)	1215 (450–2343)	1458 (753–2103)	1050 (326–2038)	0.001
	First 24 h	2770 (1135–3695)	1000 (300–2180)	1690 (838–2466)	635 (85–1360)	<0.001
	Cumulative 3 days	8900 (6853–12459)	4492 (1439–9201)	5285 (2788–7475)	1417 (255–4530)	<0.001
Blood product transfusions, mL						
	Red blood cell	450 (400–1243)	420 (370–850)	460 (384–1085)	810 (500–1233)	0.722
	Fresh frozen plasma	890 (258–1268)	695 (280–1003)	525 (290–1120)	475 (273–1105)	0.598
	Platelet concentrate	420 (115–565)	400 (128–565)	225 (128–485)	240 (180–565)	0.575
Hemoglobin (g/dL)						
	Before ECMO	11.0 ± 2.8	10.3 ± 1.8	10.7 ± 2.9	10.8 ± 2.1	0.870
	Day 1	10.8 ± 3.2	10.5 ± 3.1	11.0 ± 3.3	10.3 ± 2.0	0.782
	Day 3	10.6 ± 1.8	10.2 ± 1.6	10.3 ± 1.6	9.6 ± 1.3	0.118
Serum creatinine (mg/dL)						
	Before ECMO	1.6 ± 1.6	2.3 ± 2.6	2.0 ± 2.1	2.1 ± 2.1	0.622
	Day 1	1.8 ± 1.5	2.2 ± 1.7	2.0 ± 2.1	2.4 ± 2.0	0.595
	Day 3	1.7 ± 1.4	2.2 ± 1.6	1.9 ± 1.5	2.1 ± 1.5	0.475
AKI during ECMO (n)						
	Stage 1	14 (36.8%)	19 (50%)	16 (42.1%)	13 (34.2%)	0.588
	Stage 2	5 (13.2%)	3 (7.9%)	10 (26.3%)	13 (34.2%)	0.015
	Stage 3	4 (10.5%)	5 (13.2%)	4 (10.5%)	7 (18.4%)	0.697
	Diuretics (n)	26 (68.4%)	28 (73.7%)	26 (68.4%)	26 (68.4%)	0.941
	Inotropes (n)	32 (84.2%)	31 (81.6%)	34 (89.5%)	36 (94.7%)	0.130
	RRT (n)	17 (44.7%)	23 (60.5%)	20 (52.6%)	28 (73.7%)	0.057
	RRT initiation prior to ECMO (n)	2 (5.3%)	4 (10.5%)	4 (10.5%)	5 (13.2%)	0.272
	Time from RRT to ECMO (days)	11(2)	3 (1-18)	8 (2–14)	8 (4–25)	0.831
	RRT initiation during ECMO (n)	15 (39.5%)	19 (50%)	16 (42.1%)	23 (60.5%)	0.124
	Time from ECMO to RRT (days)	1 (0–10)	1 (0–9)	1 (0–4)	1 (0–2)	0.147
	Duration of RRT (days)	13 (4–32)	15 (7–36)	12 (6–26)	11 (4–24)	0.549
	Hospital mortality, n (%)	14 (36.8%)	18 (47.4%)	21 (55.3%)	28 (73.7%)	0.009

Data are presented as mean ± standard deviation, count, or median (interquartile range). Abbreviations: AKI, acute kidney injury; ECMO, extracorporeal membrane oxygenation; FiO_2_, fraction of inspired oxygen; PaO_2_, partial pressure of oxygen in arterial blood; PBW, predicted body weight; PEEP, positive end-expiratory pressure; RRT, renal replacement therapy; SOFA, Sequential Organ Failure Assessment; VA, venoarterial.

**Table 3 membranes-11-00567-t003:** Clinical outcomes as a function of cumulative fluid balance at 3 days after ECMO.

Variables		First Quartile	Second Quartile	Third Quartile	Fourth Quartile	*p*
		(n = 38)	(n = 38)	(n = 38)	(n = 38)	
Mortality						
	28-day hospital mortality, n (%)	9 (23.7%)	11 (28.9%)	16 (42.1%)	24 (63.2%)	<0.001
	60-day hospital mortality, n (%)	13 (34.2%)	17 (44.7%)	18 (47.4%)	25 (65.8%)	0.002
	90-day hospital mortality, n (%)	14 (36.8%)	17 (44.7%)	19 (50%)	26 (68.4%)	0.001
Other outcomes						
	Duration of ECMO (days)	8 (5–16)	12 (7–18)	10 (7–15)	7 (4–12)	0.209
	Duration of mechanical ventilator (days)	25 (11–31)	27 (14–44)	22 (15–40)	18 (10–36)	0.380
	Length of ICU stay (days)	27 (13–38)	29 (16–49)	27 (18–48)	18 (10–39)	0.451
	Length of hospital stay (days)	41 (27–61)	49 (23–73)	40 (22–69)	28 (11–53)	0.334
	ECMO-free days at day 28	15 (0–22)	9 (0–19)	8 (0–19)	0 (0–14)	0.082
	Ventilator-free days at day 28	0 (0–16)	0 (0–5)	0 (0–4)	0 (0–0)	0.002
	Ventilator-free days at day 60	32 (0–48)	14 (0–37)	0 (0–36)	0 (0–0)	0.001

Data are presented as mean ± standard deviation, count, or median (interquartile range). Abbreviations: ECMO, extracorporeal membrane oxygenation; ICU, intensive care unit.

**Table 4 membranes-11-00567-t004:** Cox proportional hazard regression models with 90-day hospital mortality as outcome.

Variables		Adjust HR (95% CI)	*p*
Model 1			
	Cumulative fluid balance from day 1 to 3 on ECMO (per 1-L increase)	1.110 (1.027–1.201)	0.009
Model 2			
	Cumulative fluid balance from day 1 to 3 on ECMO		
	First quartile	1.00 (reference)	
	Second quartile	1.200 (0.569–2.531)	0.632
	Third quartile	1.675 (0.819–3.425)	0.158
	Fourth quartile	2.710 (1.379–5.325)	0.004

Abbreviations: CI, confidence interval; ECMO, extracorporeal membrane oxygenation; FiO_2_, fraction of inspired oxygen; HR, hazard ratio; PaO_2_, partial pressure of oxygen in arterial blood. The multivariate analysis models included age, pulmonary or extrapulmonary cause of acute respiratory distress syndrome, diabetes mellitus, chronic liver disease, immunocompromised status, ECMO venoarterial mode use, mean Sequential Organ Failure Assessment score from day 1 to 3 on ECMO, mean values of ventilatory variables from day 1 to 3 on ECMO (PaO_2_/FiO_2_, positive end-expiratory pressure, peak inspiratory pressure, dynamic compliance, and mechanical power), inotropes, renal replacement therapy, cumulative total fluid input, and cumulative urine output from day 1 to 3 on ECMO. Model 1: added cumulative fluid balance from day 1 to 3 on ECMO as a continuous variable. Model 2: added cumulative fluid balance from day 1 to 3 on ECMO as categorical variables with the first quartile as the reference category.

**Table 5 membranes-11-00567-t005:** Background characteristics and clinical variables: VV-ECMO- and VA-ECMO-supported ARDS patients.

Variables		VV ECMO	VA ECMO	*p*
		(n = 128)	(n = 24)	
	Age (years)	51.2 ± 16.4	45.9 ± 15.2	0.147
	Male (gender)	89 (69.5%)	14 (58.3%)	0.281
	Body mass index (kg/m^2^)	25.8 ± 5.4	25.4 ± 4.3	0.754
ARDS etiologies				
	Pulmonary cause	103 (80.5%)	15 (62.5%)	0.053
	Extrapulmonary cause	25 (19.5%)	9 (37.5%)	0.053
	Diabetes mellitus	38 (29.7%)	2 (8.3%)	0.041
	Chronic liver disease	19 (14.8%)	2 (8.3%)	0.531
	Immunocompromised status	33 (25.8%)	7 (29.2%)	0.730
	Chronic kidney disease	16 (12.5%)	1 (4.2%)	0.313
	SOFA score before ECMO	10.6 ± 3.1	12.2 ± 3.0	0.023
	Lung injury score before ECMO	3.4 ± 0.4	3.3 ± 0.5	0.384
	PaO_2_/FiO_2_ (mmHg) before ECMO	63 (53–93)	63 (49–85)	0.787
	ECMO blood flow rate (L/min) at day 1	3.4 ± 0.7	3.4 ± 0.6	0.942
	ECMO blood flow rate (L/min) at day 3	3.9 ± 0.8	3.8 ± 0.8	0.710
	SOFA score from day 1 to day 3 on ECMO	9.6 ± 2.2	9.5 ± 3.0	0.862
	PaO_2_/FiO_2_ (mmHg) from day 1 to day 3 on ECMO	180 (135–221)	160 (106–442)	0.078
Ventilator settings from day 1 to day 3 on ECMO				
	Mechanical power (J/min)	11.5 ± 6.0	15.0 ± 6.7	0.011
	Tidal volume (mL/kg PBW)	5.8 ± 2.0	7.1 ± 2.7	0.014
	PEEP (cm H_2_O)	12.3 ± 3.3	10.4 ± 2.6	0.007
	Peak inspiratory pressure (cm H_2_O)	31.3 ± 5.3	34.0 ± 7.0	0.035
	Mean airway pressure (cm H_2_O)	17.8 ± 3.9	17.3 ± 4.4	0.592
	Dynamic compliance (mL/cm H_2_O)	19.2 ± 7.9	18.9 ± 9.1	0.870
Fluid balance, mL				
	Before ECMO 24 h	932 (−148 to 2125)	904 (−271 to 1996)	0.919
	First 24 h	1410 (53 to 2800)	1084 (78 to 2831)	0.960
	Cumulative 3 days	1301 (−873 to 3935)	833 (−894 to 4012)	0.953
Total fluid input, mL				
	Before ECMO 24 h	2938 (2243–3930)	4150 (3665–6359)	0.587
	First 24 h	4464 (3513–5828)	5005 (4398–6241)	0.525
	Cumulative 3 days	12,899 (9774–16,926)	13,425 (11,044–18,103)	0.388
Urine output, mL				
	Before ECMO 24 h	1365 (530–2350)	1857 (975–2721)	0.408
	First 24 h	1460 (396–2705)	826 (168–2264)	0.316
	Cumulative 3 days	5385 (1490–8870)	3750 (1040–7368)	0.362
Blood product transfusions, mL				
	Red blood cell	520 (393–1098)	420 (385–1306)	0.634
	Fresh frozen plasma	560 (285–1110)	490 (228–1120)	0.689
	Platelet concentrate	325 (190–560)	235 (113–545)	0.247
Hemoglobin (g/dL)				
	Before ECMO	10.7 ± 2.4	10.7 ± 2.4	0.930
	Day 1	10.6 ± 2.9	11.0 ± 3.0	0.632
	Day 3	10.1 ± 1.6	10.3 ± 1.4	0.700
Serum creatinine (mg/dL)				
	Before ECMO	2.0 ± 2.2	1.9 ± 1.5	0.715
	Day 1	2.1 ± 1.9	2.0 ± 1.0	0.662
	Day 3	2.0 ± 1.6	1.9 ± 1.0	0.928
AKI during ECMO (n)				
	Stage 1	51 (39.8%)	11 (45.8%)	0.584
	Stage 2	27 (21.1%)	4 (16.7%)	0.785
	Stage 3	15 (11.7%)	5 (20.8%)	0.225
	Diuretics (n)	91 (71.1%)	15 (62.5%)	0.400
	Inotropes (n)	110 (85.9%)	23 (95.8%)	0.311
	RRT (n)	74 (57.8%)	14 (58.3%)	0.962
	RRT initiation prior to ECMO (n)	13 (10.2%)	2 (8.3%)	1.000
	Time from RRT to ECMO (days)	5 (2–19)	8 (4)	0.720
	RRT initiation during ECMO (n)	61 (47.7%)	12 (50%)	0.833
	Time from ECMO to RRT (days)	1 (0–4)	0 (0–1)	<0.001
	Duration of ECMO (days)	10 (6–16)	8 (4–11)	0.248
	Duration of RRT (days)	13 (4–29)	10 (4–34)	0.737
	Hospital mortality, n (%)	63 (49.2%)	18 (75%)	0.020

Data are presented as mean ± standard deviation, count or median (interquartile range). Abbreviations: AKI, acute kidney injury; ARDS, acute respiratory distress syndrome; ECMO, extracorporeal membrane oxygenation; FiO_2_, fraction of inspired oxygen; PaO_2_, partial pressure of oxygen in arterial blood; PBW, predicted body weight; PEEP, positive end-expiratory pressure; RRT, renal replacement therapy; SOFA, Sequential Organ Failure Assessment; VA, venoarterial; VV, venovenous.

## Data Availability

All data will be available from the corresponding author on reasonable request.

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
