# Peer review of "Cumulative Fluid Balance during Extracorporeal Membrane Oxygenation and Mortality in Patients with Acute Respiratory Distress Syndrome"

_membranes, 2021, doi:10.3390/membranes11080567_

Round 1

Reviewer 1 Report

Thank you to the authors of the manuscript entitled” Cumulative Fluid Balance during Extracorporeal Membrane Oxygenation and Mortality in Patients with Acute Respiratory Distress Syndrome” that have put together an interesting observation.

They focused on the effect of cumulative fluid balance (CFB) during the early phase of ECMO. I think this research will lead to proper ECMO management. I have some comments.

  1. Introduction

  None.

  1. Materials and Methods

2.1. Study Design and Patients

Major comments

: This study analyzes a period of up to 10 years. Has there been any change in patient management during this period?

: Do you use a unified ECMO system (device) during this period? I think you should list the model of the ECMO system.

: The total number of populations and the number of patients excluded in each  status (1)~(5) should be stated. I recommend the explanation in the figure.

: In this study, V-V ECMO and V-A ECMO are not distinguished. The two methods have different concepts, so I think they should be unified. Isn't it better to consider 128 cases excluding V-A ECMO ?

  1. Results

 Major comments

: ECMO conditions (auxiliary flow rate, cannula size, etc.) cannot be seen. It is an important factor and should be listed. I think the average flow rate and cannula size at a certain point in time will do. There is a description of limitations, but ECMO conditions are required.

: Hemoglobin data is needed to discuss Fluid Balance. Please record it.

: Please also list the duration of ECMO in table1.

※I would like to list the above data in each table for statistical evaluation.

Please reflect in the discussion

  1. Discussion

Major comments

: I hope the discussion will be restructured based on what I have pointed out so far.

Since the hemodynamics are unstable, isn't it necessary to add volume for ECMO enforcement?  I understand that fluid overload is not good. On the other hand, you should discuss the situation where you have to put in fluid volume (patient situation, ECMO management method).

Thank you.

Author Response

Thank you to the authors of the manuscript entitled” Cumulative Fluid Balance during Extracorporeal Membrane Oxygenation and Mortality in Patients with Acute Respiratory Distress Syndrome” that have put together an interesting observation.

They focused on the effect of cumulative fluid balance (CFB) during the early phase of ECMO. I think this research will lead to proper ECMO management. I have some comments.

 Introduction

  None.

 Materials and Methods

2.1. Study Design and Patients

Major comments

Point 1: This study analyzes a period of up to 10 years. Has there been any change in patient management during this period?

Response 1:

We thank the reviewer’s comment. This study analysed a period of up to 10 years (2006-2015). The ECMO techniques were not different in our hospital during the study period (2006-2015).

Mechanical ventilation with lung-protective strategy remains the cornerstone of management strategies for acute respiratory distress syndrome (ARDS). Therefore, we added Table S1 as the Supplementary Material of the ventilator settings before and during ECMO between earlier years (2006-2011) and later years (2012-2015) and compared the differences.  

Before ECMO initiation, patients with ARDS in the earlier years (2006-2011) had significant lower PaO2/FiO2, and received significantly higher mechanical power, higher positive end-expiratory pressure (PEEP), and higher peak inspiratory pressure than patients in the later years (2012-2015) of the study period. No significant difference was observed in terms of tidal volume prior to ECMO initiation.

After ECMO support, the patients in the later years (2012-2015) received significantly lower mechanical power, lower tidal volume, higher PEEP, and lower peak inspiratory pressure than the patients in the earlier years (2006-2011). 

These findings indicated that lung-protective ventilation strategies under ECMO were applied more commonly in clinical practice to mitigate further lung injury in the later years of the study period (2012-2015). However, hospital mortality rate was not significantly different between patients in the earlier years and later years of the study period (2006-2011: 77 patients, mortality rate 54.5 %; 2012-2015: 75 patients, mortality rate 52 %, p = 0.753).

We addressed the ECMO techniques and application of lung protection principles evolved during study period in the first paragraph of Results section in the revised manuscript (3.1. Patients) as follows:

The ECMO techniques didn’t show significant difference and a unified ECMO system was used during the study period. Patients in the later years (2012-2015) received significantly lower tidal volume, higher PEEP, lower peak inspiratory pressure, and lower mechanical power during the first 3 days of ECMO than did patients in the earlier years (2006-2011) (Additional file 1: Table S1). However, hospital mortality was not significantly different between patients in the earlier years and later years of the study period (2006-2011: 77 patients, mortality rate 54.5 %; 2012-2015: 75 patients, mortality rate 52 %, p = 0.753).

We also added Table S1 as the Supplementary Material of the ventilator settings before and during ECMO between earlier years (2006-2011) and later years (2012-2015).

Point 2: Do you use a unified ECMO system (device) during this period? I think you should list the model of the ECMO system.

Response 2:

We thank the reviewer’s suggestions to address ECMO system (device) and list the model of the ECMO system evolved over the period of this study.

Despite current advances in technology and management coupled with increasing experience use of ECMO worldwide over the past decades, the ECMO techniques in our hospital were not different during the study period from 2006 to 2015, and a unified ECMO system was used (most device are Terumo model).

We didn't have a novel dual lumen venovenous (VV) ECMO cannula, the Protek Duo (Cardiac Assist, Pittsburgh, PA, USA) cannula for single cannulation with one inlet from right atrium and outlet to main pulmonary artery. Most patients with ARDS receiving ECMO mode was percutaneous VV mode with two cannula in jugular vein and femoral vein respectively, and would be replaced by venoarterial (VA) mode if encountered with cardiopulmonary cerebral resuscitation (CPCR), cardiac arrest or pulmonary hypertension. ECMO mode was changed to VV ECMO soon once the hemodynamics of patients with ARDS was stabilized.

We addressed the ECMO systems during study period in the Materials and Methods section in the revised manuscript (2.4. ECMO Systems) as follows:

The ECMO circuit consisted of a centrifugal pump and hollow-fiber microporous membrane oxygenator with heparin-bound Carmeda BioActive Surface (Carmeda, a subsidiary of WL Gore & Assoc, Flagstaff, AZ) using Capiox emergent bypass system (Terumo, Tokyo, Japan). We used two wire-wound polyurethane vascular cannula (DLP Medtronic, Minneapolis, MN [inflow, 19F to 23F, and outflow, 17F to 21F]), and the femoral-jugular venovenous (VV) ECMO was established through percutaneous cannulation. Patients would be replaced by venoarterial (VA) mode ECMO if encountered with cardiopulmonary cerebral resuscitation, cardiac arrest or pulmonary hypertension. ECMO mode was changed to VV ECMO soon once the hemodynamics of ARDS patients were stabilized

The ECMO gas flow rate was set high initially (10 L/min, pure oxygen), and the blood pump speed was gradually increased to achieve optimal oxygen saturation (90% or more). Modest volume replacement was necessary initially to improve unsteady ECMO blood flow and oxygenation. Red blood cell transfusion was performed to maintain hemoglobin at least above 10 g/dl. After optimizing the ECMO, it was crucial to remove the excessive extravascular lung water (i.e., no systemic edema, within 5% of dry weight) with diuretics or continuous renal replacement therapy to improve lung function and compliance. The hourly fluid balance goal was set at approximately -100 mL/h and modulated according to dry weight to achieve negative fluid balance, and extrapulmonary organ function was closely monitored.

We also added the statement that the ECMO techniques evolved during study period in the first paragraph of Results section in the revised manuscript (3.1. Patients) as follows:

 The ECMO techniques didn’t show significant difference and a unified ECMO system was used during the study period.

Point 3: The total number of populations and the number of patients excluded in each status (1)~(5) should be stated. I recommend the explanation in the figure.

Response 3:

We thank the reviewer’s suggestion and we added the Figure 1. The total number of population and the number of patients excluded in each status was stated and illustrated in revised Figure 1.

We added the statement in the first paragraph in the Results section (3.1. Patients) as follows:

 During the study period, a total of 192 patients with severe respiratory failure receiving ECMO were included. After excluding 40 patients, 152 patients were in the final analysis.

Point 4: In this study, V-V ECMO and V-A ECMO are not distinguished. The two methods have different concepts, so I think they should be unified. Isn't it better to consider 128 cases excluding V-A ECMO ?

Response 4:

This is an excellent point of view. We thank the reviewer to point out this problem and suggestion. We agreed that VV ECMO and VA ECMO have different concepts.

VV ECMO has become the support of choice for isolated respiratory failure in severe ARDS patients with refractory hypoxemia, and VA ECMO is less frequently applied unless there is significant associated cardiac dysfunction requiring hemodynamic support, such as unstable hemodynamic compromise, cardiogenic shock, cardiopulmonary cerebral resuscitation (CPCR), myocardial poor performance or pulmonary hypertension.

In order to compare the differences of clinical features, fluid status and outcomes of VV ECMO and VA ECMO supported ARDS patients, we add the Table 5 in the revised manuscript.

The primary aim in the current study was to investigate the effect of cumulative fluid balance during the early phase of ECMO on clinical outcomes and hospital mortality in patients with severe ARDS, and we did not observe a significant difference between VV ECMO and VA ECMO cases in terms of fluid balance before and during early phase of ECMO in our study.

As shown in Table 5, VA ECMO patients received significantly higher mechanical power, higher tidal volume, lower PEEP, and higher peak inspiratory pressure during ECMO than did VV ECMO cases (all p < 0.05), which may be associated with significantly higher hospital mortality in VA ECMO patients (p = 0.020). No significant differences were observed in terms of SOFA score, acute kidney injury development, diuretics use, blood product transfusions, inotropes support, and RRT use during ECMO.

We added paragraph to compare the differences between VV ECMO and VA ECMO supported patients in the Results section in the revised manuscript (3.6. Comparisons of VV ECMO and VA ECMO supported Patients) as follows:

As shown in Table 5, VA ECMO patients had lower percentage of diabetes mellitus. SOFA score was significantly higher in VA ECMO patients before ECMO. In terms of ventilator settings during ECMO, VA ECMO cases received significantly higher mechanical power, higher tidal volume, lower PEEP, and higher peak inspiratory pressure than did VV ECMO cases (all p < 0.05). We did not observe a significant difference between VV ECMO and VA ECMO cases in terms of fluid balance before and during ECMO. No significant differences were observed between the two groups in terms of SOFA score, acute kidney injury development, diuretics use, inotropes support, and RRT use during ECMO.VA ECMO was associated with significantly higher hospital mortality than VV ECMO (p = 0.020).

We added some description in the fifth and sixth paragraph of the Discussions section in the revised manuscript to discuss the difference of VV and VA ECMO patients (marked with red text).

  1. Results

 Major comments

Point 5: ECMO conditions (auxiliary flow rate, cannula size, etc.) cannot be seen. It is an important factor and should be listed. I think the average flow rate and cannula size at a certain point in time will do. There is a description of limitations, but ECMO conditions are required

Response 5:

This is an excellent point of view. We agreed that ECMO conditions are required and should be listed.

We addressed the ECMO systems during study period in the Materials and Methods section in the revised manuscript (2.4. ECMO Systems) that included blood flow rate, and cannula size as follows:

…We used two wire-wound polyurethane vascular cannula (DLP Medtronic, Minneapolis, MN [inflow, 19F to 23F, and outflow, 17F to 21F]), and the femoral-jugular venovenous (VV) ECMO was established through percutaneous cannulation….

…The ECMO gas flow rate was set high initially (10 L/min, pure oxygen), and the blood pump speed was gradually increased to achieve optimal oxygen saturation (90% or more). Modest volume replacement was necessary initially to improve unsteady ECMO blood flow and oxygenation…

We added ECMO condition (flow rate at day 1 and day 3 after ECMO) in Table 1, Table 2 and Table 5, and these results showed that ECMO blood flow rate was all not significantly different between survivors and nonsurvivors, among cumulative fluid balance quartiles, and between VV ECMO and VA ECMO supported patients.

Point 6: Hemoglobin data is needed to discuss Fluid Balance. Please record it.

Response 6:

We thank the reviewer’s suggestion. We added the hemoglobin data and statistical evaluation were done in Table 1, Table 2, and Table 5. The results showed that the value of hemoglobin was not significantly different between survivors and nonsurvivors, among the cumulative fluid balance quartiles, and between VV ECMO and VA ECMO supported patients.

Point 7: Please also list the duration of ECMO in table1.

Response 7:

We thank the reviewer’s suggestion and we list the duration of ECMO in Table 1. The duration of ECMO was higher in nonsurvivors than survivors, although not significant.

Point 8: ※I would like to list the above data in each table for statistical evaluation.

Please reflect in the discussion

Response 8:

 We thank the reviewer’s recommendation. The above data in each table was listed for statistical evaluation and also reflect in the discussion in the revised manuscript.

  1. Discussion

Major comments

Point 9: I hope the discussion will be restructured based on what I have pointed out so far.

Response 9:

 We thank the reviewer’s comment and the discussion was restructured based on the reviewer’s suggestions. All changes in the Discussion in the revised manuscript are marked with red text.

Point 10: Since the hemodynamics are unstable, isn't it necessary to add volume for ECMO enforcement?  I understand that fluid overload is not good. On the other hand, you should discuss the situation where you have to put in fluid volume (patient situation, ECMO management method).

Response 10:

We appreciated the reviewer’s comments and agreed that volume resuscitation is necessary for ECMO enforcement due to unstable hemodynamics during the initial phase of ECMO support.

For patients situation, most severe ARDS patients receiving VV ECMO for respiratory support, and there is no direct hemodynamic support provided by the extracorporeal circuit. The patients should be managed with fluid or blood volume replacement and inotropes as the patients who do not receive extracorporeal support, i.e., hemodynamics are managed in the same fashion as patients without VV ECMO support [1]. Bleeding remain a clinical significant issue of ECMO complication, and blood transfusions are needed to correct anemia or coagulopathy.

 For ECMO management method, a high-flow VV ECMO (3–5 L/min of extracorporeal blood flow) is required to achieve adequate blood and organs oxygenation. The blood volume should be maintained at a level high enough to keep right atrial pressure in the range of 5-10 mmHg. This will assure adequate volume for venous drainage, as long as the resistance of the drainage cannula is appropriate [2]. Echocardiography is an excellent tool to assess hemodynamic status and help guide management during VV ECMO. VA ECMO is less frequently applied in severe ARDS patients unless there is significant associated cardiac dysfunction requiring hemodynamic support. During VA ECMO support, the hemodynamics are controlled by the blood flow (pump flow plus native cardiac output), and vascular resistance. If the systemic perfusion pressure is inadequate (low urine output, poor perfusion), pressure can be increased by adding blood volume transfusion or low doses of vasopressors. If systemic oxygen delivery is not adequate (venous saturation less than 70%), increase the pump flow until perfusion is adequate. If blood volume is required to gain adequate flow, consider the relative advantages of blood product and crystalloid solution [2].

In clinical practice in our institution, the ECMO gas flow rate was set high initially (10 L/min, pure oxygen), and the blood pump speed was gradually increased to achieve optimal oxygen saturation (90% or more). Modest volume replacement was necessary initially to improve unsteady ECMO blood flow and oxygenation. After optimizing the ECMO, it was crucial to remove the excessive extravascular lung water (i.e., no systemic edema, within 5% of dry weight) with diuretics or continuous renal replacement therapy to improve lung function and compliance. The hourly fluid balance goal was set at approximately -100 mL/h and modulated according to dry weight to achieve negative fluid balance, and extrapulmonary organ function was closely monitored.

Therefore, volume replacement is usually not necessary after stabilization of the hemodynamics during early phase of ECMO with adequate fluid resuscitation and inotropes support. Fluid overload may worsen outcomes in severe ARDS patients. Besides, we closely monitored the urine output and early initiated renal replacement therapy if anuria or acidosis encountered.

We added the statement in the fourth paragraph of Materials and Methods (2.4 ECMO Systems) to describe ECMO management method as follows:

…The ECMO gas flow rate was set high initially (10 L/min, pure oxygen), and the blood pump speed was gradually increased to achieve optimal oxygen saturation (90% or more). Modest volume replacement was necessary initially to improve unsteady ECMO blood flow and oxygenation. Red blood cell transfusion was performed to maintain hemoglobin at least above 10 g/dl. After optimizing the ECMO, it was crucial to remove the excessive extravascular lung water (i.e., no systemic edema, within 5% of dry weight) with diuretics or continuous renal replacement therapy to improve lung function and compliance. The hourly fluid balance goal was set at approximately -100 mL/h and modulated according to dry weight to achieve negative fluid balance, and extrapulmonary organ function was closely monitored…

We discuss fluid volume in patient and ECMO management in the second and third paragraphs of Discussion section in the revised manuscript as follows:

Most severe ARDS patients receiving VV ECMO for respiratory support, and there is no direct hemodynamic support provided by the extracorporeal circuit. However, unstable hemodynamics was frequently occurred during the early phase of ECMO, and it is necessary to add volume for ECMO enforcement. Hemodynamics are managed with fluid or blood volume replacement and inotropes in the same fashion as patients without VV ECMO support [1]. Bleeding remain a clinical significant issue of ECMO complication, and blood transfusions are needed to correct anemia or coagulopathy.

 The blood volume during ECMO should be maintained at a level high enough to keep right atrial pressure in the range of 5-10 mmHg. This will assure adequate volume for venous drainage, as long as the resistance of the drainage cannula is appropriate. Echocardiography is an excellent tool to assess hemodynamic function and help guide management during VV ECMO. If cardiac failure occurs during VV ECMO support, the patients could be converted to VA ECMO. If the systemic perfusion pressure is inadequate (i.e., low urine output or poor perfusion), pressure can be increased by adding blood volume transfusion or low doses of vasopressors. If systemic oxygen delivery is not adequate (i.e., venous saturation less than 70%), increase the pump flow until perfusion is adequate. If blood volume is required to gain adequate flow, consider the relative advantages of blood product and crystalloid solution.

Reference:

  1. Brodie, D.; Bacchetta, M. Extracorporeal membrane oxygenation for ARDS in adults. N Engl J Med. 2011, 365, 1905-1914.
  2. Extracorporeal Life Support Organization. Available online: https://www.elso.org/Resources/Guidelines.aspx

We thank the reviewer for valuable comments. Addressing them fully has significantly strengthened the manuscript.

Reviewer 2 Report

please mention the unit of measure for CFB and the values for each quartile . please describe why the fourth quartile has the worst prognostic significance

please redo the statistics so that it is clear that the first quartile was compared to the fourth

please model the mortality in the fourth and a kaplan meier for first respectively fourth quartiles

do not forget about ethics. was the protocol approved by the institutional board how you obtained informed consent

Author Response

Point 1: please mention the unit of measure for CFB and the values for each quartile. please describe why the fourth quartile has the worst prognostic significance

Response 1:

We thank the reviewer’s suggestion. The unit of measure for cumulative fluid balance (CFB) during ECMO was ml.

Patients were stratified according to the quartiles of CFB during the first 3 days after ECMO support, and the values was as follows: quartile 1 (< -873 ml), quartile 2 (-873 to 1190 ml), quartile 3 (1190 to 3935 ml), and quartile 4 (> 3935 ml).

The primary aim in the current study was to investigate the effect of cumulative fluid balance during the first 3 days of ECMO on clinical outcomes and hospital mortality in patients with severe ARDS, and the results showed that hospital mortality was higher among patients in higher CFB quartiles. Patients in fourth quartile had highest CFB values and had worst prognostic significance.

Cox proportional hazard regression models showed that the risk of death in quartile 4 was significantly higher than in quartile 1 (Adjusted HR 2.710 [95% CI 1.379-5.325]; p = 0.004). The hazard of death of quartile 4 was also higher than quartile 2 (Adjusted HR 1.200 vs quartile 1) and quartile 3 (Adjusted HR 1.675 vs quartile 1) as with the first quartile as the reference category (Table 4).  

Kaplan-Meier 90-d survival curves also demonstrated that fourth quartile had the worst survival rate (overall comparison, p = 0.001, log-rank test). The 90-d survival rate was as follows: quartile 1 (63.2 %), quartile 2 (55.3%), quartile 3 (50%), and quartile 4 (31.6%).  

Therefore, the fourth quartile has the worst prognostic significance. These results were described in the Results of the revised manuscript (3.4. Outcomes) and presented in Table 3, Table 4, and Figure 3.

We added the unit of CFB, ml, in the section of Materials and Methods (2.2. Definitions)

We added the values for each quartile in the section of Results in the revised manuscript (3.3. Comparisons of Cumulative Fluid Balance at 3 Days after ECMO) as follows:

The values for each quartile was as follows: quartile 1 (< -873 ml), quartile 2 (-873 to 1190 ml), quartile 3 (1190 to 3935 ml), and quartile 4 (> 3935 ml).

Point 2: please redo the statistics so that it is clear that the first quartile was compared to the fourth

Response 2:

We thank the reviewer’s suggestion.

When the quartiles of CFB were considered as categorical variables with the first quartile as the reference category, the risk of death revealed a stepwise increasing trend with an increase in CFB quartile. The risk of death in quartile 4 was significantly higher than in quartile 1 (Adjusted HR 2.710 [95% CI 1.379-5.325]; p = 0.004) (Table 4).

Kaplan-Meier 90-d survival curves demonstrated that first quartile had significantly higher survival rates than quartile 4 (63.2% versus 31.6%) (p = 0.001, log-rank test) (Figure 4). The statistics was presented in Figure 4, including the comparison between first quartile and fourth quartile (Q4 vs Q1, p = 0.001). Therefore, the first quartile was compared to the fourth quartile from the above reports.

Point 3: please model the mortality in the fourth and a kaplan meier for first respectively fourth quartiles

Response 3:

We observed significantly lower overall 90-day survival rates in quartile 4 (overall comparison, p = 0.001, log-rank test), as follows: quartile 1 (63.2 %), quartile 2 (55.3%), quartile 3 (50%), and quartile 4 (31.6%).

Therefore, the 90-day mortality of each quartile was as follows:  quartile 1 (36.8%), quartile 2 (44.7%), quartile 3 (50%), and quartile 4 (68.4%).

This result was described in the Results of the revised manuscript (3.4. Outcomes) and presented in Figure 3 (Kaplan-Meier 90-d survival curves).

Point 4: do not forget about ethics. was the protocol approved by the institutional board how you obtained informed consent

Response 4:

We thank the reviewer’s recommendation. The study was conducted according to the guidelines of the Declaration of Helsinki, and the local Institutional Review Board for Human Research approved this study (CGMH IRB No. 201600632B0). The need for informed consent was waived due to the retrospective and observational nature of this study.  

The above statement was mentioned in the last sentence of the section of Materials and Methods (2.1. Study Design and Patients).

We thank the reviewer for valuable comments. Addressing them fully has significantly strengthened the manuscript.

Reviewer 3 Report

Chiu et al performed a retrospective analysis on 152 patients with severe ARDS treated with VV-ECMO, and found that cumulative fluid balance on day 3 was independently associated with hospital mortality.

Although the results of this paper are not new, as similar findings were reported by several other retrospective studies (some cited by the authors both in the introduction and the discussion), the topic is very important and this study is performed on a more large and homogeneous population.

Overall, the paper is well written, the study is well conceived and performed, and results are clearly presented and discussed.

However, the manuscript may be improved in some parts.

Major Issues:

  1. In the method section, "insensible fluid loss" is included in fluid output considered in the study. This variable is very difficul to extimate and reported formulas are usually unreliable. Which formula was used in this study? Do the authors routinely measure patients weight daily? (this could be a more reliable method to extimate large insensible fluid loss)
  2. Nonsurvivors had significantly higher SOFA score, inotropes use and RRT treatments on day 3. This could be caused by a more severe disease or inflammatory status, which may have required a more aggressive fluid resuscitation. For this reason, it could be useful to include in the analysis also total fluid input (in the paper tha authors present only cumulative fluid balance and urine output).
  3. Acute kidney injury is a major problem in critical care patients and is strongly related to fluid balance. The authors report creatinine levels, which however may be ureliable as marker of AKI during RRT. Is it possible to include in the results the number of patients with AKI (and stage according to KDIGO) and of patients on diuretics? This would be an important improvement of the manuscript.
  4. Trasfusion of blood components are very common during ECMO treatment, as bleeding is a major problem. For this reason it would be important to include in the analysis also blood product trasfusions during the first 3 days, as they could be a significant part of fluid intanke.

Minor issues:

  • Abstract: please rewrite last sentence, it seems that some part is missing
  • Table 2, Fluid balance, Cumulative 3 days: please remove commas from the numbers

Author Response

Chiu et al performed a retrospective analysis on 152 patients with severe ARDS treated with VV-ECMO, and found that cumulative fluid balance on day 3 was independently associated with hospital mortality.

Although the results of this paper are not new, as similar findings were reported by several other retrospective studies (some cited by the authors both in the introduction and the discussion), the topic is very important and this study is performed on a more large and homogeneous population.

Overall, the paper is well written, the study is well conceived and performed, and results are clearly presented and discussed.

However, the manuscript may be improved in some parts.

Major Issues:

Point 1: In the method section, "insensible fluid loss" is included in fluid output considered in the study. This variable is very difficul to extimate and reported formulas are usually unreliable. Which formula was used in this study? Do the authors routinely measure patients weight daily? (this could be a more reliable method to extimate large insensible fluid loss)

Response 1:

We thank the reviewer to point out this problem. We agreed with the reviewer that insensible fluid loss is very difficult to estimate and reported formula are usually unreliable. We also agreed that body weight could be a more reliable method to estimate large insensible fluid loss.

However, patient’s body weight was measured at ICU admission and then weekly, not daily, in our hospital in clinical practice. The primary aim of this study was to investigate the effect of cumulative fluid balance during the first 3 days of ECMO on clinical outcomes and hospital mortality in severe ARDS patients. Because of the retrospective nature of our analysis, body weight was not recorded daily, and we used the formula to approximately estimated insensible fluid loss.

The insensible fluid loss was routinely estimated in the ICUs in our institution every 8 hours daily and the formula was as follows:

10 mL/kg/day + 2.5 mL/kg/day per degree centigrade above 37 °C (max body weight in equation 100 kg) (× 0.6 if intubated) [1, 2].

References:

  1. Cox, P. Insensible water loss and its assessment in adult patients: a review. Acta Anaesthesiol Scand. 1987, 31, 771-6.
  2. Hessels, L.; Oude, Lansink-Hartgring, A.; Zeillemaker-Hoekstra, M.; Nijsten, M.W. Estimation of sodium and chloride storage in critically ill patients: a balance study. Ann Intensive Care. 2018, 8, 97.

Point 2: Nonsurvivors had significantly higher SOFA score, inotropes use and RRT treatments on day 3. This could be caused by a more severe disease or inflammatory status, which may have required a more aggressive fluid resuscitation. For this reason, it could be useful to include in the analysis also total fluid input (in the paper tha authors present only cumulative fluid balance and urine output).

Response 2:

This is an excellent point of view. We thank the reviewer to point out this problem and suggestion.

 We add total fluid input in Table 1, and the result showed that total fluid input was higher in nonsurvivors than survivors, although not significantly different. We also add total fluid input in Table 2 and the data showed no significantly different among cumulative fluid balance quartiles.

  Cox proportional hazard regression models were performed again and the data were the same as the previous report, which was shown in Table 4.

Point 3: Acute kidney injury is a major problem in critical care patients and is strongly related to fluid balance. The authors report creatinine levels, which however may be ureliable as marker of AKI during RRT. Is it possible to include in the results the number of patients with AKI (and stage according to KDIGO) and of patients on diuretics? This would be an important improvement of the manuscript.

Response 3:

We appreciate the reviewer’s comment to point out these issues. The numbers of patients with AKI (stage by KDIGO) and patients on diuretics were added in Table 1 and Table 2.  

Although the development of AKI was higher in nonsurvivors, but the results was not significantly different (Table 1). The development of AKI and diuretic use were also not significantly different among cumulative fluid balance quartiles, except AKI stage 2 (Table 2).

 We added the definition of acute kidney injury in the Materials and Methods section in the revised manuscript (2.2. Definitions) as follows:

Acute kidney injury was defined according to the KDIGO (Kidney Disease: Improving Global Outcomes) classification system and staged for severity based on serum creatinine or urine output [1].

Reference:

  1. Kellum, J.A.; Lameire, N. KDIGO AKI Guideline Work Group. Diagnosis, evaluation, and management of acute kidney injury: a KDIGO summary (Part 1). Crit Care. 2013, 17, 204.

Point 4: Trasfusion of blood components are very common during ECMO treatment, as bleeding is a major problem. For this reason it would be important to include in the analysis also blood product trasfusions during the first 3 days, as they could be a significant part of fluid intanke

Response 4:

We thank the reviewer’s comment and agreed that blood transfusion is a significant part of fluid intake during ECMO.

 We analysed the blood product transfusions, including red blood cell, fresh frozen plasma, and platelet concentrate. We added the values in Table 1, and Table 2. The results showed that blood product transfusions (red blood cell, fresh frozen plasma, and platelet concentrate) were all not significantly different between survivors and nonsurvivors, and among cumulative fluid balance quartiles.

Minor issues:

Point 5: Abstract: please rewrite last sentence, it seems that some part is missing

Response 5:

 We thank the reviewer to point out the inappropriate description in the last sentence in the Abstract.  We revised the statement as follows:

 Our findings indicate a conservative treatment approach to avoid fluid overload during the early phase of ECMO when dealing with severe ARDS patients.

Point 6: Table 2, Fluid balance, Cumulative 3 days: please remove commas from the numbers

Response 6:

We thank the reviewer to point out this problem. We remove commas from the numbers in Table 2, Fluid balance, Cumulative 3 days.

We thank the reviewer for valuable comments. Addressing them fully has significantly strengthened the manuscript.

Round 2

Reviewer 1 Report

The author is responsive to reviewer's comments. This  paper has been improved.

Reviewer 2 Report

The authors did some significant efforts to improve the quality of their manuscript.

Reviewer 3 Report

The authors provided a complete point-by-point response to all questions and the manuscript has been greatly improved.

I have no other issues. 

I wish to the authors the best success for this paper and future research.